# Technology-Enabled Senior Living: A Preliminary Report on Stakeholder Perspectives

**DOI:** 10.3390/healthcare12030381

**Published:** 2024-02-01

**Authors:** Vera Stara, Elvira Maranesi, Johanna Möller, Cecilia Palmier, Toshimi Ogawa, Ryan Browne, Marine Luc, Rainer Wieching, Jerome Boudy, Roberta Bevilacqua

**Affiliations:** 1Scientific Direction, IRCCS INRCA, 60124 Ancona, Italy; v.stara@inrca.it (V.S.); r.bevilacqua@inrca.it (R.B.); 2Diocesan Caritas Association for the Archdiocese of Cologne, D-50668 Cologne, Germany; johanna.moeller@caritasnet.de; 3Service de Gériatrie 1&2, AP-HP, Hôpital Broca, 75013 Paris, France; cecilia.palmier@aphp.fr; 4Smart-Aging Research Center, Tohoku University, Sendai 980-8575, Japan; toshimi.ogawa.e6@tohoku.ac.jp (T.O.); browne.ryan.e2@tohoku.ac.jp (R.B.); 5AGE Platform Europe, 1150 Brussels, Belgium; marine.luc@age-platform.eu; 6Institute for New Media and Information Systems, University Siegen, D-57072 Siegen, Germany; rainer.wieching@uni-siegen.de; 7Institut Mines-Telecom, Telecom SudParis, SAMOVAR IP Paris, 91011 Evry, France; jerome.boudy@telecom-sudparis.eu

**Keywords:** active and healthy ageing, stakeholders, smart living solutions, older adults, qualitative method

## Abstract

Background: The integration of stakeholders is crucial in developing smart living technologies to support the autonomy of elderly populations. Despite the clear benefits of these technologies, there remains a significant gap in comprehensive research. Methods: This study presents the viewpoints of 19 stakeholders from Europe and Japan, focusing on the sustainability of smart living solutions for Active and Healthy Ageing (AHA). Data were gathered through qualitative semi-structured interviews and analysed using a Framework Analysis approach. Results: Analysis of the interviews revealed six key sustainability categories: addressing the unmet needs of older adults, functionalities of the smart living coach, integration within organizations, identified barriers, financial considerations, and the social role of the smart living coach. Conclusions: This research underscores the importance of evaluating user needs through the involvement of various stakeholders, including the elderly, their caregivers, professionals, technicians, service providers, and government bodies. Collaborative efforts are essential to generate new evidence demonstrating the value of smart living solutions in facilitating Active and Healthy Ageing.

## 1. Introduction

Stakeholder engagement in designing health services seems to be a path to improve patient access and outcomes and patient-centred care [1,2,3,4]. This engagement remains a key factor when co-creating smart living solutions to sustain the independent living of older adults, since stakeholders are individuals or groups responsible for or are affected by health and healthcare-related decisions [5,6,7]. Due to demographic change, a low number of care professionals, and changes in family structures, there is now a turn towards digital solutions to support care and prevention. Innovative approaches can aid older people in managing their health and well-being [8,9,10,11,12,13]. However, this is only possible if several conditions are met: end-users are consulted, they possess sufficient ICT literacy, and they can access affordable and interoperable solutions that facilitate the collaboration between older adults, their informal and professional health caregivers, community members, and policy-makers [14]. Stakeholder engagement is thus crucial to ensure the sustainability, acceptance, and uptake of digital solutions. Therefore, there is a need to establish cooperative agreements between all actors involved in the design of innovative tools to build reasonable support systems that are sustainable because they can fulfil the real needs of older adults [15,16,17]. The potential of smart living solutions is evident, but there is still a need for thorough investigations to fully realise the potential of smart living for older adults in regions with ageing populations, such as in Europe and Japan. There are, indeed, studies that confirm the expectation that technology can help community services to improve care, quality of life, well-being of communities, and reduce the cost of care for older adults [18,19,20,21]. Most of these studies have confirmed that technology has become user-friendly, comfortable, and acceptable to users. Also, they have pointed out that, as a group, pre-frail older adults are considered the best fit for these initiatives.

Based on this complex scenario, the present study shares data collected during the e-VITA project (EU-Japan Virtual Coach for Smart Ageing), which is aimed at improving the well-being of older adults in Europe and Japan, thereby promoting active and healthy ageing, contributing to independent living, and reducing the risks of social exclusion of older adults using a personalized virtual coach capable of interacting with its user. The e-VITA virtual coach, consisting of social robots, will thus provide personalized recommendations and interventions for sustainable well-being in a smart living environment at home.

The design and development of the e-VITA system followed a user-centred and value-sensitive participatory design approach, enrolling older end-users [22], clinicians [14], and stakeholders across Europe and Japan. In this study, the following types of stakeholders were invited and then involved: communities and municipalities, welfare organizations and NGOs, health insurance companies and volunteering markets. This wide range of stakeholders is focused on allowing different areas of daily living and the well-being of EU and Japanese senior citizens to be covered. Furthermore, some of the stakeholders provide content for combined real coach services, which may be associated with a virtual coach system on demand. This is crucial as seniors are a very heterogeneous group and demand services and products that can enhance the growing industry sector of smart living solutions. This paper aims to report the perspectives of 19 stakeholders across Europe and Japan to discuss the sustainability of smart living solutions for AHA in Europe and Japan.

## 2. Materials and Methods

An essential phase of the e-VITA project is the acquisition of a detailed picture of attitudes, drivers, barriers, and opportunities as useful information for the future impact of the system through the consultation of relevant stakeholders across Europe and Japan. The data collected will be used to assess the level of interest for e-VITA and explore the public health, ethical, technological, and organizational use and socio-economic conditions for integrating solutions in the chain of health and homecare services.

To do this, we developed an English language qualitative semi-structured interview guideline and then translated it into the languages of the countries doing the interviews. The interview guide is shown in Figure 1. It is composed of seven questions covering the following areas of interest: unmet needs of older adults, functionalities of the coach, integration in organization, barriers, finances, and social role of the coach.

The data were collected in March/April 2021.

### 2.1. Recruitment

An invitation email was sent to invite the stakeholders. A researcher called those persons who accepted to be a part of this study and explained the objectives of the project, as well as the aim of the survey. An information letter was attached to the email, and the stakeholders were kindly invited to visit the project website [23] as a repository of all the significant content, videos, podcasts, press releases, and materials related to the project activities. Interviews with stakeholders in Germany, Italy, and France were performed via video call due to COVID-19 restrictions. For the same reason, a self-written questionnaire was used in Japan. All of the interviews were recorded, and then a transcription was made. The details of the participants across Europe and Japan are reported in Table 1.

### 2.2. Qualitative Analysis

Data were collected and analysed in the native language of each site (German, Italian, French, and Japanese) and then the local results were translated into English by a third person and combined cross-nationally. The cross-national findings were analysed using the Framework Analysis Method [24,25]. The MAXQDA software package (MAXQDA 2022, VERBI Software GmbH, Berlin, Germany) was used for the qualitative research. To ensure comparability between the countries, the main categories were created deductively based on the topics of the interviews. Six categories were established: unmet needs of older adults, functionalities of the coach, integration in organization, barriers, finances, and social role of the coach. The substantive differentiation of these topics was carried out inductively to reflect the original observations of the respondents. Then, the lead author (V.S.) coded to validate the data analysis. The investigator triangulation method [26] was applied using two other researchers from the team (R.B., E.M.) to validate the list of codes, minimize personal bias, ensure validity and reliability [27], and meet trustworthiness criteria [28,29]. Following this coding process, quotations from the same topics were grouped under a common code. Specific quotations were reported to clarify the meaning of the codes emerging from the analysis.

## 3. Results

### 3.1. Unmet Needs

Respondents have shared different unmet needs. Stakeholders (IT01, IT02, GE03, FR03) reported that the COVID-19 pandemic brought digital and technological inequalities to the surface. The introduction of lockdowns in the spring of 2020 increased the use of the Internet, online communication, and electronic social services. Initially, there were very few people who knew how to move into this new digital world, especially among older adults, but this emergency brought the opportunity to bring them closer to the world of technology.

From this perspective, a robot that provides information and procedures would speed the process of digital inclusion; therefore, e-Vita could be an effective solution to counteract the increased digital divide after the COVID-19 crisis. For example, such a robot could be an effective tool to help older people perform practical tasks that they cannot do independently.

Currently, digital devices are used in all aspects of life, including health and social services, to reduce territorial inequalities regarding health or social conditions, to fight against isolation, and to offer advice and support.

“*Initially, very few persons knew how to move into the digital world, but the emergency also brought a benefit, and so far, many of our older adults are using these tools better. Consider, for example, the Digital Identity service: many older people do not know what it is and have not yet achieved it. This impedes requesting bonuses or making the most of online public utility services. It is indeed an unmet need.*” (IT01)

The digital divide risk represents a fracture with the outside world for seniors over 65 (IT01, IT03, FR01), but a way to counteract the digital divide could be the promotion of e-health literacy; the need to receive information and to communicate, regardless of the level of education and digital skills, is greater for seniors (IT01, IT03).

“*The fundamental need of the older age group is not so much to have knowledge of the technological tool itself, but education in its use. Therefore, a robot made available to them would have to facilitate connection with the outside world, first, with their family doctor.*” (IT03)

This connection would also open new possibilities for the emancipation of older people to make choices and live independently in their environment (FR03, GE03): “*I think this need for emancipation is fundamental: older adults want to make choices, decide to stay at home in complete safety*” (FR03).

### 3.2. Functionalities of the Coach

The promotion of a healthy lifestyle (IT01, IT02, GE02, GE03, FR02, FR05, FR06, JP01, JP03–07) was the most quoted function among stakeholders. This promotion covers different aspects, from booking medical appointments, pharmaceuticals, and general reminders, to making meaningful recommendations. Based on the knowledge that the coach has about a person, the coach should be able to make emphatic recommendations on the behaviour of the users by informing, suggesting, and communicating about daily life aids to stay healthy.

“*He would have to meet me empathically somehow like that. It should motivate me to deal positively with my ageing, to be in a good positive mood and to bring me sunshine in the gravy everyday life of old age.*” (GE03)

The coach should give key messages for healthy living based on physical activity, diet, sleep, meeting other people, and intellectual learning by offering functionalities for each dimension of ageing well since the target of e-VITA are populations under 75 years old with decreased vitality or depressive symptoms (anxiety and worry).

“*The aim of such functionalities should be to simplify the user. How do we hide the complexity for the user? How do we make his life easier? And if we’re talking about making it easier to age well, we have to make it easier to live with nutrition, physical activity, social links and organisation.*” (FR02)

The functionality of matching information was raised among French stakeholders (FR01, FR04, FR05) who considered all the innovations made by private or public organisations in their countries. Indeed, there are currently many things created in terms of prevention, but these innovations are scattered throughout the territory, and people do not know what exists or how to take advantage of them.

“*We could imagine that the older person asks to do gymnastics, and the virtual coach suggests the closest gym with the address, telephone number and map to direct them. The coach should rely on the territory’s offer to give precise information.*” (FR05)

A connection with other services such as telemedicine and home environments was seen as important by some stakeholders (IT03, JP04, JP05, and JP07), for example, linking the system with a telemedicine service for “*measuring some basic health parameters, currently, for example, an important data as of oxygen saturation in the blood*” (IT03), or to communicate possible cognitive and physical disorders to a medical professional: “*the coach will serve to link the information afterwards to the medical profession, nurse or home care assistant to take the decision or help the elderly person to accept a solution*” (FR01).

Integration with the home environment using SNS could also be a key factor for Japanese stakeholders who imagined introducing such systems to comprehensive support centres, home care support offices, long-term care insurance sections in their municipality, neighbourhood associations, welfare commissioners, and projects of preventive long-term care.

Not only connection with other services or the home environment, but also connection through social avenues was mentioned (FR04 and FR05); a virtual coach could connect users with the outside world by proposing social activities or contacting family members if requested.

Japanese and Italian respondents underlined that interaction with the system should be simple for all since older adults are unfamiliar with innovations and could have negative images of a virtual coach (JP01, JP02, JP04, JP06, and JP07). Japanese specialists (JP01, JP02, and JP05–07) suggested that a lower IT literacy of users and care staff would be the biggest barrier to e-VITA. It would be useful to instruct them on how to benefit from technology in the community (JP04, JP05, and JP07).

For that reason, usability and ease of use of the system are mandatory: “*The commands must be simple and intuitive; this, in general, is a fundamental condition: few buttons, almost all functions activated by voice*” (IT03).

Data protection and preventing loss through unauthorised access were topics reported only in Germany (GE01, GE02). Stakeholders underlined the importance of monitoring this function when approaching virtual coaching for the older population.

Lastly, GE01 suggested emotional recognition as a primary function of the coach. The system should be able to know about the emotions of the users and act accordingly: “*He would have to feel my mood, like what I need today, what is good for my soul*” (GE01).

### 3.3. Integration in Organization

Only Italian (IT01, IT02, IT03) and German stakeholders (GE03) reported thoughts about integrating the e-VITA system into their organizations.

The system would be definitely a plus for any organization. For example, “*it should be linked to our social activities and courses and would help the association’s work, as well as perhaps promoting a correct lifestyle and physical activity or mobility. Alternatively, it could help us with our home food delivery service, receiving the list of products from the older person, then transmitting the order. It could be the tool connecting our organization to the members. For example, with the function of calling numbers of our offices, by voice command, or the call to use our home transport service, the robot could memorize the bookings of visits or health check-ups, calculating the frequency, for instance, that that older person turns to our accompanying service once a week. Considering e-VITA as a very advanced technological system, for medical use, the robot could process health records/profiles of each older person who turns to our association*” (IT03).

### 3.4. Barriers

The problem of privacy and data security was raised in all of the European countries and in Japan. The sensitivity of an individual’s private data being collected and recorded by digital tools raised concerns about the privacy of this information.

“*What happens with the data must be transparent to the user. If the user is uncertain about the usage, he/she might choose not to use it.*” (GE03)

“*By now, even older users are more used to dealing with this issue; they are more aware that some data are necessarily shared, and if e-VITA is guaranteed by a regional or municipal public institution, they will trust it.*” (IT01)

Another barrier detected was digital literacy (IT01, IT 03, GE03, JP02, JP04, JP05, JP06). Even if the engagement of older adults in approaching digital technology is higher than before, poor familiarity with such tools is still the main issue. For that reason, it is very important to support older adults in managing new devices and design tools developed with user-centred principles.

“*In the experience of the NGO, older adults are not always keen on learning something new. Therefore, it would be helpful if the coach was not entirely new but had already well-known elements within this group. Another problem the NGO has faced in their daily work is the sensitivity of touchscreens. They are not always entirely adapted to the skin of older adults, and sometimes they can have problems interacting with the screen. The NGO proposed using voice interfaces instead to make a two-way interface (voice and touch).*” (GE03)

The issue of trustworthiness (trust in the accuracy of the information shared by the system) was reported only in Germany (GE01, GE02), whereas the issue of accessibility of internet connection (IT01, IT02) and aesthetics (IT03) was only reported in Italy.

“*The system does not have an aspect that is too human, which could frighten the older person: a neutral form, pleasant for sure, but it must also be manageable; for example, I can imagine that it has a horizontal support surface on which to rely, it should be robust, thus giving a feeling of solidity in case of need.*” (IT03)

### 3.5. Finances

The topic of finances is seen by participants as something difficult to estimate or to solve in terms of assistance. Many options were mentioned, but they really depended on national regulations, regional actions, and personal opportunities. GE01 summarized the point: “*So that’s always the big question, who pays for it? When you’re on the road in a preventive context. It’s always difficult because no one wants to pay for it. The insurance companies don’t want to, and the patients don’t want to. Or the other way around, the health insurance companies might pay for it if they say it brings something. That is if there is a benefit. But yes, it is difficult*”. JP04 suggested that it is necessary to consider whether it is covered by a self-financed service or long-term care insurance. Long-term care insurance in Japan is available for 10% to 30% of the cost to the individual. In addition, assuming that it is a service package using telecommunication, both the initial and running costs are considered to be required. In this case, he believes the model would work well with cost-sharing, like subscriptions. Regardless of whether insurance is applied, in terms of the burden on the individual user, JP04 suggested that the cost should be around JPY 10,000 to 20,000 per month. In Europe, participants agreed that a virtual coach must have a relatively low price or one that can be adjusted according to income and be easily accessible through contacts with local authorities.

### 3.6. Social Role of the Coach

Participants have argued two different roles for a digital coach: a careful professional and a facilitator. The coach could be seen as being midway between a doctor and a friend: a point of reference, a sort of ‘virtual person’ or consultant you trust, capable of listening to the special needs of the older person, avoiding a too-anthropomorphic shape.

“*It should not have large dimensions because it would give more the feeling of being controlled or invaded by technology than of a tool that serves to help. A more symbolic, geometric form, perhaps leading back to an imaginary linked to health, would be appropriate as a kind of information point to address health: it can also have a face or a more human visage in the design, but without excess.*” (IT01)

FR03 took a contrary position: “*if the coach gives information, I would say we don’t have the impression that it’s a fake but he must not behave like a doctor*”. The coach could also be a medical assistant associated with a medical team.

For IT03, it would have a “bridge” role with all the digital potential and services, particularly health ones, but not be limited to being a kind of facilitator. In addition, even more informal and friendly aspects are important, allowing for a bit of sociality. To talk to each other, to ask questions and listen to the answers with a dialogue based on the interests of the older person, such as, for example, getting information on the time of their favourite TV show.

## 4. Discussion

Within this paper, the perspectives of 19 stakeholders across Europe and Japan were reported to analyse the sustainability of smart living solutions for AHA in Europe and Japan. The cross-national findings were analysed using the Framework Analysis Method [30,31], and six categories were built: unmet needs of older adults, functionalities of the coach, integration in the organization, barriers, finances, and social role of the coach. The qualitative findings point out relevant strengths and weaknesses in the sustainability of smart living solutions for AHA. Stakeholders agreed that technology can inform, suggest, and communicate daily life aids to stay healthy; for example, by proposing physical or cognitive exercises and social participations. Effectively, this can be seen as a strength that is in line with large clinical studies [30,31,32,33,34] that demonstrated the extent to which technology can provide benefits in multiple domains, such as physical activity, cognitive stimulation, and participation in social life. Promoting and informing healthy lifestyles could also help older adults learn about their health and cope with the ageing process. For the stakeholders involved in this study, another strength offered by technology is related to the possibility of facilitating a connection with the outside world or to link with doctors. This point is also mapped in other different studies. Indeed, digital tools could enhance and enrich the lives of older adults by facilitating better interpersonal relationships with loved ones, friends, and the community, and so mitigate loneliness and isolation. This social engagement could then reinforce the sense of belonging as a social member of the community and impact active and healthy ageing in place [35,36,37,38,39,40].

Despite these strengths, different weaknesses were detected. Across Europe and Japan, all the stakeholders recognized issues with data security, the digital divide, e-health literacy, and a lack of funding to support ownership of smart technology. Technology can collect passive and active data from users to improve care, provide daily assistance, and enhance quality of life, but at the same time, these tools introduce challenging privacy and security risks. Previous work has extensively discussed these privacy and security challenges [41]. Therefore, data privacy and data security are integral to the successful implementation of such technologies. The digital divide and low e-health literacy are seen as other weak points. According to a report from Internet World Statistics, in 2023, the number of global Internet users was over 5.3 billion, with an Internet penetration rate of 67.9% [42]. Currently, the constant innovation and upgrading of medical service models generates new methods for requesting health information, enabling health communication; the COVID-19 pandemic has since made the importance of digital health more prominent [43,44,45]. This important change was underlined by different stakeholders involved in this study. The lack of digital health literacy and the digital divide have become the main obstacles hindering older adults from participating in digital society and accessing digital health services. They can be seen also as barriers to entering the market.

Finally, finances are another perceived barrier. In Japan, considering the cost of the e-VITA system, it would be desirable to use long-term care insurance or long-term care prevention instead of medical insurance. It is possible to make the system one of the services provided by private companies. On the other hand, from a European perspective, ageing strategies and tools exist but are approached at a regional level. The European Union developed and made available funding schemes and monitoring tools and encouraged the exchange of good practices for active and healthy ageing, but because social and health policies are out of the remit of its competencies, only EU member states are able to legislate or provide support for a system such as e-VITA. In the case of e-VITA, efforts to find structural support could maybe be focused at the local or regional levels, rather than the national level. The national level, however, could be solicited to address the topic of healthy ageing and solutions to support healthy ageing, and thus generate or increase the general public’s interest about it.

### 4.1. Comparison with Older Adults’ Perspective

In another study [22], we collected and identified needs, perceptions, and possible barriers expressed by 58 adults aged 65 and over in France, Italy, Germany, and Japan. All the recruited participants were regular PC and/or smartphone users. During the pandemic, they increasingly learned how to use technology to stay in contact with relatives and acquaintances, indicating a general openness to technology as a measure of social interactions. Some participants were also avid users of self-tracking devices or apps. The participants in all settings saw the potential benefits of a virtual coach for health-related advice reminders or recommendation features. With regard to the concrete, practical use of a virtual coach, the respondents imagined that social contacts could be established and maintained through the system. Opinions differed on the idea that a virtual coach could be a social or emotional support.

It is also important to mention that the concern about a coach not being trustworthy enough and too commanding among older adults was frequently stated in all countries. Participants want to preserve a sense of control over the device. A vital worry was privacy. Users expressed an intention to determine which personal data was used and transferred. Moreover, participants were worried about potential financial barriers and the cost of the system. Thus, with regards to a market solution, one has to manage the trade-offs of satisfying diverse needs and expectations while creating low-cost solutions.

### 4.2. Limitations

Even though this study benefitted from including stakeholders across Europe and Japan, it had limitations. Data was collected in the national languages of Italy, France, Germany, and Japan, and then the data were translated into English. This language switch and the national restrictions could have introduced biases that do not allow for the generalization of the results.

We collected data during the pandemic, which may have significantly altered the impressions expressed by the participants. Two years after the COVID-19 period, technology has quite improved, so we plan to compare these perspectives retrospectively, performing another round of interviews at the beginning of 2024.

Finally, according to the consolidated criteria for reporting qualitative research [46], since the researchers’ position about the participants and the research topic was not addressed, this could be assumed as another bias. Furthermore, a mixed-methods approach could have guaranteed a broad understanding of thoughts. These limitations can be used as seeds for future research.

## 5. Conclusions

Whitin this study, the sustainability of smart living solutions for AHA in Europe and Japan is discussed by analysing the direct voice of 19 stakeholders. Different strengths and weaknesses are mapped. Even if the stakeholders are interested parties, a comprehensive analysis of the currently on-going initiatives and studies for the ageing population is challenging. Assessing user needs is a fundamental task that needs the participation of different actors: from senior citizens, their formal and informal caregivers and professionals, to technicians, service providers, communities and municipalities, as well as governments (payers) that need to collaborate to create new evidence for the value of smart solutions to enable AHA.

## Figures and Tables

**Figure 1 healthcare-12-00381-f001:**
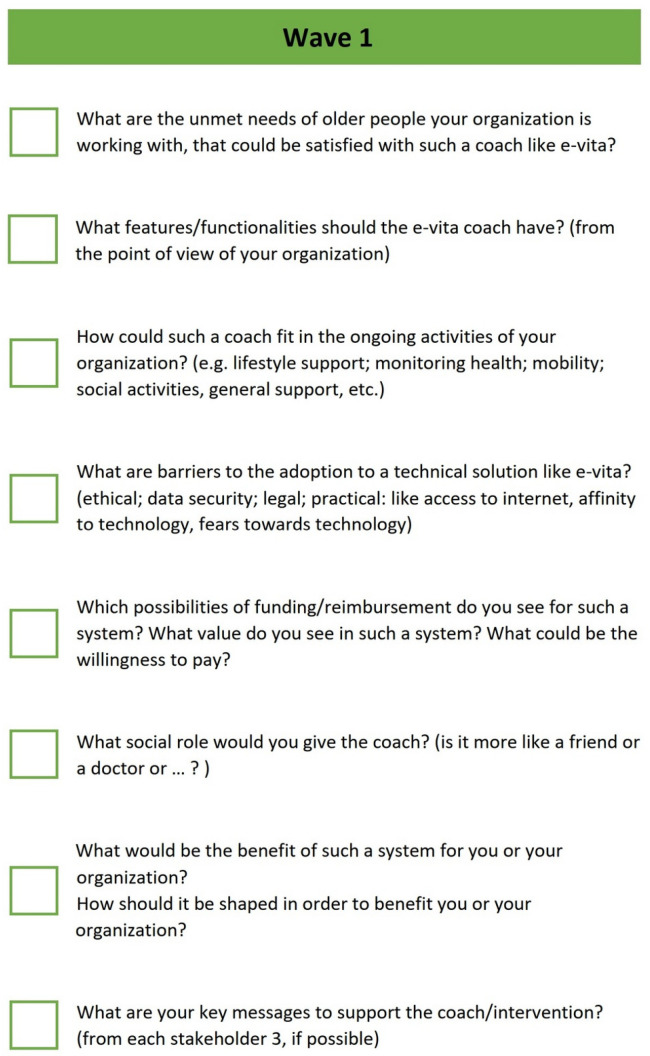
The interview guide.

**Table 1 healthcare-12-00381-t001:** Participants across Europe and Japan.

Code	Country	Expertise
IT01	Italy	Teacher at the University of the Third Age of Ancona
IT02	Italy	Regional Secretary Marche region Union for pensioners
IT03	Italy	President of a non-profit association
FR01	France	Health insurance company and innovation projects
FR02	France	Health insurance company and innovation projects
FR03	France	Geriatrician
FR04	France	Social actor at the Municipality of Paris
FR05	France	Social actor at the Municipality of Paris
FR06	France	Employee at an organization that promotes ageing well in retirement
GE01	Germany	Employee at an IT company–working with health insurance and developing e-health software for them
GE02	Germany	Employee at an IT company–working with health insurance and developing e-health software for them
GE03	Germany	Employee at an NGO focused on technology adaptation
JP01	Japan	Social expert
JP02	Japan	Social expert
JP03	Japan	Politician
JP04	Japan	Social expert
JP05	Japan	Social expert
JP06	Japan	Technician
JP07	Japan	Economist

## Data Availability

The data presented in this study are available in the article itself.

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
