# Peer review of "Technology-Enabled Senior Living: A Preliminary Report on Stakeholder Perspectives"

_healthcare, 2024, doi:10.3390/healthcare12030381_

Round 1

Reviewer 1 Report

Comments and Suggestions for Authors

Reviewed on 26th Dec 2023

·        Title could be brief and precise eg Technology enabled senior living- Stakeholders perspective. A preliminary Report  

·        Complex long sentences can be made shorter. It appears to the reviewer that the authors are not very familiar with scientific English. The text itself is 13 pages with 9000 plus words. It will be difficult to sustain the interest of the average reader. The length of the article could be reduced by shortening sentences, being precise, focused and avoiding repetitions

·        It is stated “ ---then translated into the languages of the countries doing the interviews” The exact languages need to be mentioned. Ideally the translated version should have been re translated by at third person independently back to English. ? was this done

·        It is well known that education, age, gender, socio economic status, digital literacy, presence or absence of caregivers etc etc play a major role in the beneficiary’s perspective . There is a total absence of this information

·         For the international reader a paragraph is essential on what are the options for a senior citizen in those countries ? who pays the bill is there insurance coverage etc etc

·        Conceding that stakeholder’s perspective is important the reviewer is of the opinion that interviewing 3/ 3/6/7   individuals

         from four different countries 3 years ago and extrapolating observations made at the height of the pandemic in March/April   

          2021.  to  inferences worthy of publication 3 years later may be questionable.

·         During the last 3 years there has also been considerable change in technology and perception of end users have also changed

·        It is suggested that the study be repeated with larger numbers. Comparison of views in 2024 to views in 2021would also be useful and add to the value of the paper

Comments on the Quality of English Language

Reviewed on 26th Dec 2023

·        Title could be brief and precise eg Technology enabled senior living- Stakeholders perspective. A preliminary Report  

·        Complex long sentences can be made shorter. It appears to the reviewer that the authors are not very familiar with scientific English. The text itself is 13 pages with 9000 plus words. It will be difficult to sustain the interest of the average reader. The length of the article could be reduced by shortening sentences, being precise, focused and avoiding repetitions

·        It is stated “ ---then translated into the languages of the countries doing the interviews” The exact languages need to be mentioned. Ideally the translated version should have been re translated by at third person independently back to English. ? was this done

·        It is well known that education, age, gender, socio economic status, digital literacy, presence or absence of caregivers etc etc play a major role in the beneficiary’s perspective . There is a total absence of this information

·         For the international reader a paragraph is essential on what are the options for a senior citizen in those countries ? who pays the bill is there insurance coverage etc etc

·        Conceding that stakeholder’s perspective is important the reviewer is of the opinion that interviewing 3/ 3/6/7   individuals

         from four different countries 3 years ago and extrapolating observations made at the height of the pandemic in March/April   

          2021.  to  inferences worthy of publication 3 years later may be questionable.

·         During the last 3 years there has also been considerable change in technology and perception of end users have also changed

·        It is suggested that the study be repeated with larger numbers. Comparison of views in 2024 to views in 2021would also be useful and add to the value of the paper

Author Response

C1: Title could be brief and precise eg Technology enabled senior living- Stakeholders perspective. A preliminary Report 

A1: Thank you for this suggestion. Now the title is “Technology enabled senior living: A preliminary Report on Stakeholders perspective”

C2: Complex long sentences can be made shorter. It appears to the reviewer that the authors are not very familiar with scientific English. The text itself is 13 pages with 9000 plus words. It will be difficult to sustain the interest of the average reader. The length of the article could be reduced by shortening sentences, being precise, focused and avoiding repetitions

A2: We revised the manuscript accordingly to these points.

C3: It is stated “ ---then translated into the languages of the countries doing the interviews” The exact languages need to be mentioned. Ideally the translated version should have been re translated by at third person independently back to English? was this done

A3: We have clarified this important point  in 2.2. Statistical Analysis. We confirmed that a third person re-translated back to English the result

C4: It is well known that education, age, gender, socio economic status, digital literacy, presence or absence of caregivers etc etc play a major role in the beneficiary’s perspective . There is a total absence of this information

A4: We agree that education, age, gender, socio economic status, digital literacy, presence or absence of caregivers etc etc play a major role in the beneficiary’s perspective but the focus of this study was to discover and report the perspectives of stakeholders across Europe and Japan to discuss the sustainability of smart living solutions for AHA in Europe and Japan.

C5: For the international reader a paragraph is essential on what are the options for a senior citizen in those countries? who pays the bill is there insurance coverage etc etc

A5: Since there is a publication related to the perspective of older adults (ref 22: Möller, J.; Bevilacqua, R.; Browne, R.; Shinada, T.; Dacunha, S.; Palmier, C.; et al. User Perceptions and Needs Analysis of a Virtual Coach for Active and Healthy Ageing—An International Qualitative Study. International journal of environmental research and public health, 2022, 19(16), 10341.) we did not report data about senior citizen in the countries involved to avoid any problems of self-plagiarism.

C6: Conceding that stakeholder’s perspective is important the reviewer is of the opinion that interviewing 3/ 3/6/7   individuals from four different countries 3 years ago and extrapolating observations made at the height of the pandemic in March/April 2021.  to  inferences worthy of publication 3 years later may be questionable.

C7: During the last 3 years there has also been considerable change in technology and perception of end users have also changed

C8: It is suggested that the study be repeated with larger numbers. Comparison of views in 2024 to views in 2021would also be useful and add to the value of the paper

A6-A7-A8: We agree with the reviewer that our data could be questionable for the timing of collection, but our aim is to compare these perspectives in the long term. At the beginning of 2024, we planned another round of interviews to check (if any) changing views. We add this info to the discussion.

Reviewer 2 Report

Comments and Suggestions for Authors

Thank you for your very interesting study. I enjoyed reading it and I wish you every success in this important work.

I have made a few suggestions that may help demonstrate the level of rigour in your study. They relate primarily to the method used, reflexivity, and the presentation of your results.

In the recruitment section, you wrote that a self-written questionnaire was used. As surveys are generally considered quantitative (because the data translate to numbers), it would be useful to confirm if your survey tool included open-ended questions that were subsequently analysed in the same manner as the interview transcripts. If your survey produced quantitative data, then this is a mixed methods study and would need to be written up as such.

In qualitative research, and in alignment with COREQ (Consolidated criteria for reporting qualitative research), the position of the researcher/s in relation to the participants and the research topic, should be addressed, usually by each researcher undertaking a reflexivity exercise. The issues normally considered include (amongst other things) the assumptions and biases of the researcher, and how these were managed (or bracketed) during the data collection and analysis phases. Can you clarify whether or not the researcher/s considered their position? Did the researcher/s have experience/training in qualitative interviewing prior to gathering data?

As the transcripts become the de facto raw data, it is vital that they are correct. Were the transcripts verified in their original language, by someone other than the transcriber? Who verified the subsequent translations?

You stated that you used a framework analysis approach, which is appropriate.

The results section seemed to be primarily consisting of quotes. Generally, quotes are used more sparingly, and are for illustrative purposes only, i.e. the text should still make sense if the quotes were deleted. This is not the case here. I suggest that the results section is re-written to increase the text and reduce the quotes and this may, in turn, reflect the deeper meaning of the data (where appropriate), and that these meanings can then be illustrated using brief quotations. Re-writing will facilitate the reader as the current presentation includes very dense text that will make it difficult for the reader to extract meaning.

Each quote includes a range of concepts, so I think it may be wise to revisit the analysis or which part of the quote you wish to present, in order to illustrate your point.

The way many of the quotes are presented would suggest that individuals made a statement, and this was interpreted as a subcode, rather than a theme. This would suggest that you did not find themes across all the interviews. What is not clear, is whether there was a consensus around any of these statements by your participants (later in the discussion there was some comment related to this, but it would sit well here in the results). If there was a consensus around a concept, across the interviews, then I suggest you could present the concept as a theme, and in this case, there would be a statement as to the collective perception or belief about the concept, without attribution, and this could be followed by a quote that may or may not be attributed (currently, there is a debate about whether attribution is needed). In some instances, you clarify that there is agreement across stakeholders. This suggests that the other statements are isolated opinions. If this is the case, then it would be important to clarify the point, as this would not be a thematic analysis as suggested by Braun and Clarke (reference 25). There should also be more clarity around the levels of consensus (or not) relating to the different concepts.

This first section refers to the ‘digital divide’, ‘territorial inequalities’ and ‘emancipation of older people’. Based solely on the data you have presented, these do not reflect the title of the theme: Unmet need, and I suggest that you consider something more reflective of the content.

Discussion: This section included a useful summary of the results, and the strengths of the work. The weaknesses referred to the intervention (e-VITA) and not to the actual study. I suggest the limitations of the study are included after the ‘strengths’ section. This could include the comments regarding translation and biases that are presented further below, and might also include issues around data saturation. The final section about the implications was well written and relevant.

Comments on the Quality of English Language

The work would benefit from further proof-reading as there are some minor errors in spelling and grammar.

Author Response

Thank you for your very interesting study. I enjoyed reading it and I wish you every success in this important work.

I have made a few suggestions that may help demonstrate the level of rigour in your study. They relate primarily to the method used, reflexivity, and the presentation of your results.

C1: In the recruitment section, you wrote that a self-written questionnaire was used. As surveys are generally considered quantitative (because the data translate to numbers), it would be useful to confirm if your survey tool included open-ended questions that were subsequently analysed in the same manner as the interview transcripts. If your survey produced quantitative data, then this is a mixed methods study and would need to be written up as such.

A1: Thank you for this question. We process data only considering qualitative information without translating to numbers. We declared it as a limitation of the study.

C2: In qualitative research, and in alignment with COREQ (Consolidated criteria for reporting qualitative research), the position of the researcher/s in relation to the participants and the research topic, should be addressed, usually by each researcher undertaking a reflexivity exercise. The issues normally considered include (amongst other things) the assumptions and biases of the researcher, and how these were managed (or bracketed) during the data collection and analysis phases. Can you clarify whether or not the researcher/s considered their position? Did the researcher/s have experience/training in qualitative interviewing prior to gathering data?

A2: Thank you for raising this point. We declared it as a limitation of the study.

C3: As the transcripts become the de facto raw data, it is vital that they are correct. Were the transcripts verified in their original language, by someone other than the transcriber? Who verified the subsequent translations?

A3: We clarified this point in Subsection 2.2: “Data were collected and analyzed in the native language of each site (German, Italian, French and Japanese)and then the local results were translated into English language  by a third person and combined cross-nationally.”

C4: You stated that you used a framework analysis approach, which is appropriate.

A4: Thank you

C5: The results section seemed to be primarily consisting of quotes. Generally, quotes are used more sparingly, and are for illustrative purposes only, i.e. the text should still make sense if the quotes were deleted. This is not the case here. I suggest that the results section is re-written to increase the text and reduce the quotes and this may, in turn, reflect the deeper meaning of the data (where appropriate), and that these meanings can then be illustrated using brief quotations. Re-writing will facilitate the reader as the current presentation includes very dense text that will make it difficult for the reader to extract meaning.

A5: Thank you for suggesting us to completely re-write the Result. Please check the new version.

C6: Each quote includes a range of concepts, so I think it may be wise to revisit the analysis or which part of the quote you wish to present, in order to illustrate your point.

A6: Done

C7: The way many of the quotes are presented would suggest that individuals made a statement, and this was interpreted as a subcode, rather than a theme. This would suggest that you did not find themes across all the interviews. What is not clear, is whether there was a consensus around any of these statements by your participants (later in the discussion there was some comment related to this, but it would sit well here in the results). If there was a consensus around a concept, across the interviews, then I suggest you could present the concept as a theme, and in this case, there would be a statement as to the collective perception or belief about the concept, without attribution, and this could be followed by a quote that may or may not be attributed (currently, there is a debate about whether attribution is needed). In some instances, you clarify that there is agreement across stakeholders. This suggests that the other statements are isolated opinions. If this is the case, then it would be important to clarify the point, as this would not be a thematic analysis as suggested by Braun and Clarke (reference 25). There should also be more clarity around the levels of consensus (or not) relating to the different concepts.

 A7: We hope that the new version of the Resul clarify our method as stated in 2.2. Statistical Analysis

C8: This first section refers to the ‘digital divide’, ‘territorial inequalities’ and ‘emancipation of older people’. Based solely on the data you have presented, these do not reflect the title of the theme: Unmet need, and I suggest that you consider something more reflective of the content.

A8: Thank you, please check the new title (also in accordance with the other reviewer who raised the same comment).

C9: Discussion: This section included a useful summary of the results, and the strengths of the work. The weaknesses referred to the intervention (e-VITA) and not to the actual study. I suggest the limitations of the study are included after the ‘strengths’ section. This could include the comments regarding translation and biases that are presented further below, and might also include issues around data saturation. The final section about the implications was well written and relevant.

A9: Thank you for this suggestion. Please check the new subsection 4.1 Limitations.

Round 2

Reviewer 1 Report

Comments and Suggestions for Authors

Comments on the Quality of English Language

Complex long sentences can be made shorter. It appears to the reviewer that the authors are not very familiar with scientific English. The text itself is 13 pages with 9000 plus words. It will be difficult to sustain the interest of the average reader. The length of the article could be reduced by shortening sentences, being precise, focused and avoiding repetitions

Author Response

We would like to thank you for your helpful comments and suggestions. We very much appreciate the time and effort invested in the review of our manuscript.

Please check the new version of the manuscript. The text (without references) is now ten pages, less than 5000 words. A mother tongue author of the authorship proof read the content. Moreover, a new sub paragraph has been added (4.1) aimed to describe the comparison with older adults’ perspective.

Reviewer 2 Report

Comments and Suggestions for Authors

Thank you for all your hard work on this paper, and for your responses to my comments.

The paper reads well, and I wish you every success with the next round of interviews as you develop this research.

Comments on the Quality of English Language

There are a few minor editing issues to be addressed, but overll, this is much improved.

Author Response

We would like to thank you for your helpful comments and suggestions. We very much appreciate the time and effort invested in the review of our manuscript.